# RANDOMIZED AUTOMATIC DIFFERENTIATION

**Deniz Oktay,**[1] **Nick McGreivy,**[2] **Joshua Aduol,**[1] **Alex Beatson,**[1] **Ryan P. Adams**[1]
Princeton University
Princeton, NJ
`{doktay,mcgreivy,jaduol,abeatson,rpa}@princeton.edu`

## ABSTRACT

The successes of deep learning, variational inference, and many other fields have been aided by specialized implementations of reverse-mode automatic differentiation (AD) to compute gradients of mega-dimensional objectives. The AD techniques underlying these tools were designed to compute exact gradients to numerical precision, but modern machine learning models are almost always trained with stochastic gradient descent. Why spend computation and memory on exact (minibatch) gradients only to use them for stochastic optimization? We develop a general framework and approach for randomized automatic differentiation (RAD), which can allow unbiased gradient estimates to be computed with reduced memory in return for variance. We examine limitations of the general approach, and argue that we must leverage problem specific structure to realize benefits. We develop RAD techniques for a variety of simple neural network architectures, and show that for a fixed memory budget, RAD converges in fewer iterations than using a small batch size for feedforward networks, and in a similar number for recurrent networks. We also show that RAD can be applied to scientific computing, and use it to develop a low-memory stochastic gradient method for optimizing the control parameters of a linear reaction-diffusion PDE representing a fission reactor.

## 1 INTRODUCTION

Deep neural networks have taken center stage as a powerful way to construct and train massively-parametric machine learning (ML) models for supervised, unsupervised, and reinforcement learning tasks. There are many reasons for the resurgence of neural networks—large data sets, GPU numerical computing, technical insights into overparameterization, and more—but one major factor has been the development of tools for automatic differentiation (AD) of deep architectures. Tools like PyTorch and TensorFlow provide a computational substrate for rapidly exploring a wide variety of differentiable architectures without performing tedious and error-prone gradient derivations. The flexibility of these tools has enabled a revolution in AI research, but the underlying ideas for reverse-mode AD go back decades. While tools like PyTorch and TensorFlow have received huge dividends from a half-century of AD research, they are also burdened by the baggage of design decisions made in a different computational landscape. The research on AD that led to these ubiquitous deep learning frameworks is focused on the computation of Jacobians that are *exact* up to numerical precision. However, in modern workflows these Jacobians are used for *stochastic* optimization. We ask:

> *Why spend resources on exact gradients when we're going to use stochastic optimization?*

This question is motivated by the surprising realization over the past decade that deep neural network training can be performed almost entirely with first-order stochastic optimization. In fact, empirical evidence supports the hypothesis that the regularizing effect of gradient noise *assists* model generalization (Keskar et al., 2017; Smith & Le, 2018; Hochreiter & Schmidhuber, 1997). Stochastic gradient descent variants such as AdaGrad (Duchi et al., 2011) and Adam (Kingma & Ba, 2015) form the core of almost all successful optimization techniques for these models, using small subsets of the data to form the noisy gradient estimates.

---

[1] Department of Computer Science
[2] Department of Astrophysical Sciences

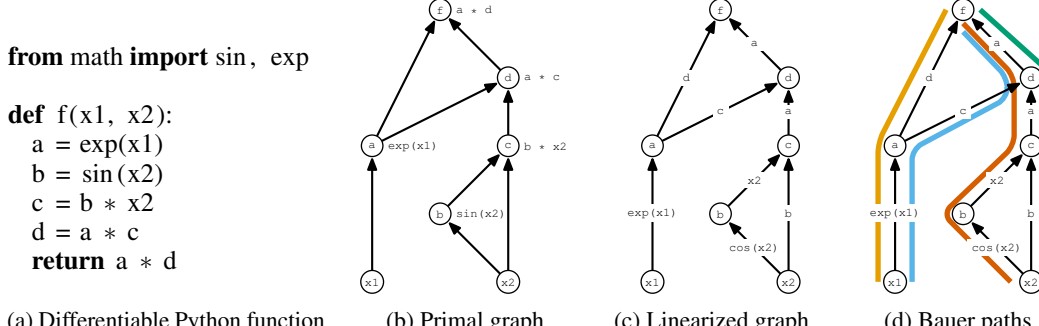

```
from math import sin, exp

def f(x1, x2):
    a = exp(x1)
    b = sin(x2)
    c = b * x2
    d = a * c
    return a * d
```

(a) Differentiable Python function    (b) Primal graph    (c) Linearized graph    (d) Bauer paths

Figure 1: Illustration of the basic concepts of the linearized computational graph and Bauer's formula. (a) a simple Python function with intermediate variables; (b) the primal computational graph, a DAG with variables as vertices and flow moving upwards to the output; (c) the linearized computational graph (LCG) in which the edges are labeled with the values of the local derivatives; (d) illustration of the four paths that must be evaluated to compute the Jacobian. (Example from Paul D. Hovland.)

The goals and assumptions of automatic differentiation as performed in classical and modern systems are mismatched with those required by stochastic optimization. Traditional AD computes the derivative or Jacobian of a function accurately to numerical precision. This accuracy is required for many problems in applied mathematics which AD has served, e.g., solving systems of differential equations. But in stochastic optimization we can make do with inaccurate gradients, as long as our estimator is unbiased and has reasonable variance. We ask the same question that motivates mini-batch SGD: why compute an exact gradient if we can get noisy estimates cheaply? By thinking of this question in the context of AD, we can go beyond mini-batch SGD to more general schemes for developing cheap gradient estimators: in this paper, we focus on developing gradient estimators with low memory cost. Although previous research has investigated approximations in the forward or reverse pass of neural networks to reduce computational requirements, here we replace deterministic AD with *randomized* automatic differentiation (RAD), trading off of computation for variance *inside* AD routines when imprecise gradient estimates are tolerable, while retaining unbiasedness.

## 2 AUTOMATIC DIFFERENTIATION

Automatic (or *algorithmic*) differentiation is a family of techniques for taking a program that computes a differentiable function $f : \mathbb{R}^n \to \mathbb{R}^m$, and producing another program that computes the associated derivatives; most often the Jacobian: $\mathcal{J}[f] = f' : \mathbb{R}^n \to \mathbb{R}^{m \times n}$. (For a comprehensive treatment of AD, see Griewank & Walther (2008); for an ML-focused review see Baydin et al. (2018).) In most machine learning applications, $f$ is a loss function that produces a scalar output, i.e., $m = 1$, for which the gradient with respect to parameters is desired. AD techniques are contrasted with the method of *finite differences*, which approximates derivatives numerically using a small but non-zero step size, and also distinguished from *symbolic differentiation* in which a mathematical expression is processed using standard rules to produce another mathematical expression, although Elliott (2018) argues that the distinction is simply whether or not it is the compiler that manipulates the symbols.

There are a variety of approaches to AD: source-code transformation (e.g., Bischof et al. (1992); Hascoet & Pascual (2013); van Merrienboer et al. (2018)), execution tracing (e.g., Walther & Griewank (2009); Maclaurin et al.), manipulation of explicit computational graphs (e.g., Abadi et al. (2016); Bergstra et al. (2010)), and category-theoretic transformations (Elliott, 2018). AD implementations exist for many different host languages, although they vary in the extent to which they take advantage of native programming patterns, control flow, and language features. Regardless of whether it is constructed at compile-time, run-time, or via an embedded domain-specific language, all AD approaches can be understood as manipulating the *linearized computational graph* (LCG) to collapse out intermediate variables. Figure 1 shows the LCG for a simple example. These computational graphs are always directed acyclic graphs (DAGs) with vertices as variables.

Let the outputs of $f$ be $y_j$, the inputs $\theta_i$, and the intermediates $z_l$. AD can be framed as the computation of a partial derivative as a sum over all paths through the LCG DAG (Bauer, 1974):

$$\frac{\partial y_j}{\partial \theta_i} = \mathcal{J}_\theta[f]_{j,i} = \sum_{[i \to j]} \prod_{(k,l) \in [i \to j]} \frac{\partial z_l}{\partial z_k} \tag{1}$$

where $[i \to j]$ indexes paths from vertex $i$ to vertex $j$ and $(k, l) \in [i \to j]$ denotes the set of edges in that path. See Figure 1d for an illustration. Although general, this naïve sum over paths does not take advantage of the structure of the problem and so, as in other kinds of graph computations, dynamic programming (DP) provides a better approach. DP collapses substructures of the graph until it becomes bipartite and the remaining edges from inputs to outputs represent exactly the entries of the Jacobian matrix. This is referred to as the *Jacobian accumulation problem* (Naumann, 2004) and there are a variety of ways to manipulate the graph, including vertex, edge, and face elimination (Griewank & Naumann, 2002). Forward-mode AD and reverse-mode AD (backpropagation) are special cases of more general dynamic programming strategies to perform this summation; determination of the optimal accumulation schedule is unfortunately NP-complete (Naumann, 2008).

While the above formulation in which each variable is a scalar can represent any computational graph, it can lead to structures that are difficult to reason about. Often we prefer to manipulate vectors and matrices, and we can instead let each intermediate $z_l$ represent a $d_l$ dimensional vector. In this case, $\partial z_l / \partial z_k \in \mathbb{R}^{d_l \times d_k}$ represents the intermediate Jacobian of the operation $z_k \to z_l$. Note that Equation 1 now expresses the Jacobian of $f$ as a sum over chained matrix products.

## 3 Randomizing Automatic Differentiation

We introduce techniques that could be used to decrease the resource requirements of AD when used for stochastic optimization. We focus on functions with a scalar output where we are interested in the gradient of the output with respect to some parameters, $\mathcal{J}_\theta[f]$. Reverse-mode AD efficiently calculates $\mathcal{J}_\theta[f]$, but requires the full linearized computational graph to either be stored during the forward pass, or to be recomputed during the backward pass using intermediate variables recorded during the forward pass. For large computational graphs this could provide a large memory burden.

The most common technique for reducing the memory requirements of AD is gradient checkpointing (Griewank & Walther, 2000; Chen et al., 2016), which saves memory by adding extra forward pass computations. Checkpointing is effective when the number of "layers" in a computation graph is much larger than the memory required at each layer. We take a different approach; we instead aim to save memory by increasing gradient variance, without extra forward computation.

Our main idea is to consider an unbiased estimator $\hat{\mathcal{J}}_\theta[f]$ such that $\mathbb{E}\hat{\mathcal{J}}_\theta[f] = \mathcal{J}_\theta[f]$ which allows us to save memory required for reverse-mode AD. Our approach is to determine a sparse (but random) linearized computational graph during the forward pass such that reverse-mode AD applied on the sparse graph yields an unbiased estimate of the true gradient. Note that the original computational graph is used for the forward pass, and randomization is used to determine a LCG to use for the backward pass in place of the original computation graph. We may then decrease memory costs by storing the sparse LCG directly or storing intermediate variables required to compute the sparse LCG.

In this section we provide general recipes for randomizing AD by sparsifying the LCG. In sections 4 and 5 we apply these recipes to develop specific algorithms for neural networks and linear PDEs which achieve concrete memory savings.

### 3.1 Path Sampling

Observe that in Bauer's formula each Jacobian entry is expressed as a sum over paths in the LCG. A simple strategy is to sample paths uniformly at random from the computation graph, and form a Monte Carlo estimate of Equation 1. Naïvely this could take multiple passes through the graph. However, multiple paths can be sampled without significant computation overhead by performing a topological sort of the vertices and iterating through vertices, sampling multiple outgoing edges for each. We provide a proof and detailed algorithm in the appendix. Dynamic programming methods such as reverse-mode automatic differentiation can then be applied to the sparsified LCG.

## 3.2 RANDOM MATRIX INJECTION

In computation graphs consisting of vector operations, the vectorized computation graph is a more compact representation. We introduce an alternative view on sampling paths in this case. A single path in the vectorized computation graph represents many paths in the underlying scalar computation graph. As an example, Figure 2c is a vector representation for Figure 2b. For this example,

$$\frac{\partial y}{\partial \theta} = \frac{\partial y}{\partial C}\frac{\partial C}{\partial B}\frac{\partial B}{\partial A}\frac{\partial A}{\partial \theta} \tag{2}$$

where $A, B, C$ are vectors with entries $a_i, b_i, c_i$, $\partial C/\partial B$, $\partial B/\partial A$ are $3 \times 3$ Jacobian matrices for the intermediate operations, $\partial y/\partial C$ is $1 \times 3$, and $\partial A/\partial \theta$ is $3 \times 1$.

We now note that the contribution of the path $p = \theta \rightarrow a_1 \rightarrow b_2 \rightarrow c_2 \rightarrow y$ to the gradient is,

$$\frac{\partial y}{\partial C}P_2\frac{\partial C}{\partial B}P_2\frac{\partial B}{\partial A}P_1\frac{\partial A}{\partial \theta} \tag{3}$$

where $P_i = e_i e_i^T$ (outer product of standard basis vectors). Sampling from $\{P_1, P_2, P_3\}$ and right multiplying a Jacobian is equivalent to sampling the paths passing through a vertex in the scalar graph.

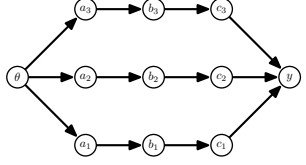

(a) Independent Paths Graph

In general, if we have transition $B \rightarrow C$ in a vectorized computational graph, where $B \in \mathbb{R}^d, C \in \mathbb{R}^m$, we can insert a random matrix $P = d/k \sum_{s=1}^{k} P_s$ where each $P_s$ is sampled uniformly from $\{P_1, P_2, \ldots, P_d\}$. With this construction, $\mathbb{E}P = I_d$, so

$$\mathbb{E}\left[\frac{\partial C}{\partial B}P\right] = \frac{\partial C}{\partial B}. \tag{4}$$

If we have a matrix chain product, we can use the fact that the expectation of a product of independent random variables is equal to the product of their expectations, so drawing independent random matrices $P_B, P_C$ would give

$$\mathbb{E}\left[\frac{\partial y}{\partial C}P_C\frac{\partial C}{\partial B}P_B\right] = \frac{\partial y}{\partial C}\mathbb{E}\left[P_C\right]\frac{\partial C}{\partial B}\mathbb{E}\left[P_B\right] = \frac{\partial y}{\partial C}\frac{\partial C}{\partial B} \tag{5}$$

Right multiplication by $P$ may be achieved by sampling the intermediate Jacobian: one does not need to actually assemble and multiply the two matrices. For clarity we adopt the notation $\mathcal{S}_P\left[\partial C/\partial B\right] = \partial C/\partial B P$. This is sampling (with replacement) $k$ out of the $d$ vertices represented by $B$, and only considering paths that pass from those vertices.

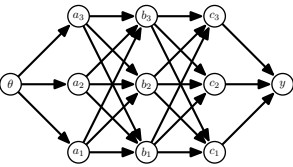

(b) Fully Interleaved Graph

(c) Vector graph for (b).

Figure 2: Common computational graph patterns. The graphs may be arbitrarily deep and wide. (a) A small number of independent paths. Path sampling has constant variance with depth. (b) The number of paths increases exponentially with depth; path sampling gives high variance. Independent paths are common when a loss decomposes over data. Fully interleaved graphs are common with vector operations.

The important properties of $P$ that enable memory savings with an unbiased approximation are

$$\mathbb{E}P = I_d \qquad \text{and} \qquad P = RR^T, R \in \mathbb{R}^{d \times k}, k < d. \tag{6}$$

We could therefore consider other matrices with the same properties. In our additional experiments in the appendix, we also let $R$ be a random projection matrix of independent Rademacher random variables, a construction common in compressed sensing and randomized dimensionality reduction.

In vectorized computational graphs, we can imagine a two-level sampling scheme. We can both sample paths from the computational graph where each vertex on the path corresponds to a vector. We can also sample within each vector path, with sampling performed via matrix injection as above.

In many situations the full intermediate Jacobian for a vector operation is unreasonable to store. Consider the operation $B \rightarrow C$ where $B, C \in \mathbb{R}^d$. The Jacobian is $d \times d$. Thankfully many common operations are element-wise, leading to a diagonal Jacobian that can be stored as a $d$-vector. Another common operation is matrix-vector products. Consider $Ab = c$, $\partial c/\partial b = A$. Although $A$ has many more entries than $c$ or $b$, in many applications $A$ is either a parameter to be optimized or is easily recomputed. Therefore in our implementations, we do not directly construct and sparsify the Jacobians. We instead sparsify the input vectors or the compact version of the Jacobian in a way that has the same effect. Unfortunately, there are some practical operations such as softmax that do not have a compactly-representable Jacobian and for which this is not possible.

### 3.3 VARIANCE

The variance incurred by path sampling and random matrix injection will depend on the structure of the LCG. We present two extremes in Figure 2. In Figure 2a, each path is independent and there are a small number of paths. If we sample a fixed fraction of all paths, variance will be constant in the depth of the graph. In contrast, in Figure 2b, the paths overlap, and the number of paths increases exponentially with depth. Sampling a fixed fraction of all paths would require almost all edges in the graph, and sampling a fixed fraction of vertices at each layer (using random matrix injection, as an example) would lead to exponentially increasing variance with depth.

It is thus difficult to apply sampling schemes without knowledge of the underlying graph. Indeed, our initial efforts to apply random matrix injection schemes to neural network graphs resulted in variance exponential with depth of the network, which prevented stochastic optimization from converging. We develop tailored sampling strategies for computation graphs corresponding to problems of common interest, exploiting properties of these graphs to avoid the exploding variance problem.

## 4 CASE STUDY: NEURAL NETWORKS

We consider neural networks composed of fully connected layers, convolution layers, ReLU nonlinearities, and pooling layers. We take advantage of the important property that many of the intermediate Jacobians can be compactly stored, and the memory required during reverse-mode is often bottlenecked by a few operations. We draw a vectorized computational graph for a typical simple neural network in figure 3. Although the diagram depicts a dataset of size of 3, mini-batch size of size 1, and 2 hidden layers, we assume the dataset size is $N$. Our analysis is valid for any number of hidden layers, and also recurrent networks. We are interested in the gradients $\partial y/\partial W_1$ and $\partial y/\partial W_2$.

### 4.1 MINIBATCH SGD AS RANDOMIZED AD

At first look, the diagram has a very similar pattern to that of 2a, so that path sampling would be a good fit. Indeed, we could sample $B < N$ paths from $W_1$ to $y$, and also $B$ paths from $W_2$ to $y$. Each path corresponds to processing a different mini-batch element, and the computations are independent.

In empirical risk minimization, the final loss function is an average of the loss over data points. Therefore, the intermediate partials $\partial y/\partial h_{2,x}$ for each data point $x$ will be independent of the other data points. As a result, if the same paths are chosen in path sampling for $W_1$ and $W_2$, and if we are only interested in the stochastic gradient (and not the full function evaluation), the computation graph only needs to be evaluated for the data points corresponding to the sampled paths. This exactly corresponds to mini-batching. The paths are visually depicted in Figure 3b.

### 4.2 ALTERNATIVE SGD SCHEMES WITH RANDOMIZED AD

We wish to use our principles to derive a randomization scheme that can be used on top of mini-batch SGD. We ensure our estimator is unbiased as we randomize by applying random matrix injection independently to various intermediate Jacobians. Consider a path corresponding to data point 1. The contribution to the gradient $\partial y/\partial W_1$ is

$$\frac{\partial y}{\partial h_{2,1}} \frac{\partial h_{2,1}}{\partial a_{1,1}} \frac{\partial a_{1,1}}{\partial h_{1,1}} \frac{\partial h_{1,1}}{\partial W_1} \tag{7}$$

Using random matrix injection to sample every Jacobian would lead to exploding variance. Instead, we analyze each term to see which are memory bottlenecks.

$\partial y/\partial h_{2,1}$ is the Jacobian with respect to (typically) the loss. Memory requirements for this Jacobian are independent of depth of the network. The dimension of the classifier is usually smaller $(10 - 1000)$ than the other layers (which can have dimension $10,000$ or more in convolutional networks). Therefore, the Jacobian at the output layer is not a memory bottleneck.

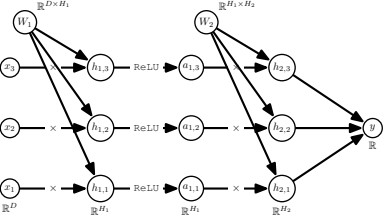

(a) Neural network computational graph

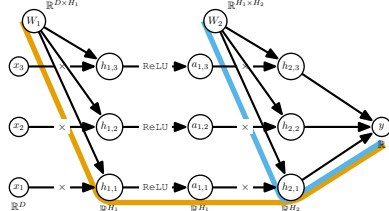

(b) Computational Graph with Mini-batching

Figure 3: NN computation graphs.

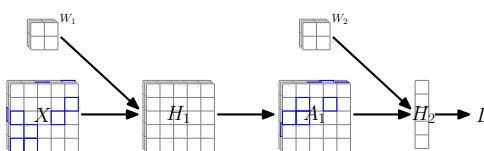

Figure 4: Convnet activation sampling for one mini-batch element. $X$ is the image, $H$ is the pre-activation, and $A$ is the activation. $A$ is the output of a ReLU, so we can store the Jacobian $\partial A_1/\partial H_1$ with 1 bit per entry. For $X$ and $H$ we sample spatial elements and compute the Jacobians $\partial H_1/\partial W_1$ and $\partial H_2/\partial W_2$ with the sparse tensors.

$\partial h_{2,1}/\partial a_{1,1}$ is the Jacobian of the hidden layer with respect to the previous layer activation. This can be constructed from $W_2$, which must be stored in memory, with memory cost independent of mini-batch size. In convnets, due to weight sharing, the effective dimensionality is much smaller than $H_1 \times H_2$. In recurrent networks, it is shared across timesteps. Therefore, these are not a memory bottleneck.

$\partial a_{1,1}/\partial h_{1,1}$ contains the Jacobian of the ReLU activation function. This can be compactly stored using 1-bit per entry, as the gradient can only be $1$ or $0$. Note that this is true for ReLU activations in particular, and not true for general activation functions, although ReLU is widely used in deep learning. For ReLU activations, these partials are not a memory bottleneck.

$\partial h_{1,1}/\partial W_1$ contains the memory bottleneck for typical ReLU neural networks. This is the Jacobian of the hidden layer output with respect to $W_1$, which, in a multi-layer perceptron, is equal to $x_1$. For $B$ data points, this is a $B \times D$ dimensional matrix.

Accordingly, we choose to sample $\partial h_{1,1}/\partial W_1$, replacing the matrix chain with $\frac{\partial y}{\partial h_{2,1}} \frac{\partial h_{2,1}}{\partial a_{1,1}} \frac{\partial a_{1,1}}{\partial h_{1,1}} \mathcal{S}_{P_{W_1}} \left[ \frac{\partial h_{1,1}}{\partial W_1} \right]$. For an arbitrarily deep NN, this can be generalized:

$$\frac{\partial y}{\partial h_{d,1}} \frac{\partial h_{d,1}}{\partial a_{d-1,1}} \frac{\partial a_{d-1,1}}{\partial h_{d-1,1}} \cdots \frac{\partial a_{1,1}}{\partial h_{1,1}} \mathcal{S}_{P_{W_1}} \left[ \frac{\partial h_{1,1}}{\partial W_1} \right], \quad \frac{\partial y}{\partial h_{d,1}} \frac{\partial h_{d,1}}{\partial a_{d-1,1}} \frac{\partial a_{d-1,1}}{\partial h_{d-1,1}} \cdots \frac{\partial a_{2,1}}{\partial h_{2,1}} \mathcal{S}_{P_{W_2}} \left[ \frac{\partial h_{2,1}}{\partial W_2} \right]$$

This can be interpreted as sampling activations on the backward pass. This is our proposed alternative SGD scheme for neural networks: along with sampling data points, we can also sample activations, while maintaining an unbiased approximation to the gradient. This does not lead to exploding variance, as along any path from a given neural network parameter to the loss, the sampling operation is only applied to a single Jacobian. Sampling for convolutional networks is visualized in Figure 4.

## 4.3 NEURAL NETWORK EXPERIMENTS

We evaluate our proposed RAD method on two feedforward architectures: a small fully connected network trained on MNIST, and a small convolutional network trained on CIFAR-10. We also evaluate our method on an RNN trained on Sequential-MNIST. The exact architectures and the calculations for the associated memory savings from our method are available in the appendix. In Figure 5 we include empirical analysis of gradient noise caused by RAD vs mini-batching.

We are mainly interested in the following question:

*For a fixed memory budget and fixed number of gradient descent iterations, how quickly does our proposed method optimize the training loss compared to standard SGD with a smaller mini-batch?*

Reducing the mini-batch size will also reduce computational costs, while RAD will only reduce memory costs. Theoretically our method could reduce computational costs slightly, but this is not our focus. We only consider the memory/gradient variance tradeoff while avoiding adding significant overhead on top of vanilla reverse-mode (as is the case for checkpointing).

Results are shown in Figure 6. Our feedforward network full-memory baseline is trained with a mini-batch size of 150. For RAD we keep a mini-batch size of 150, and try 2 different configurations. For "same sample", we sample with replacement a 0.1 fraction of activations, and the same activations are sampled for each mini-batch element. For "different sample", we sample a 0.1 fraction of activations, independently for each mini-batch element. Our "reduced batch" experiment is trained without RAD with a mini-batch size of 20 for CIFAR-10 and 22 for MNIST. This achieves similar memory budget as RAD with mini-batch size 150. Details of this calculation and of hyperparameters are in the appendix.

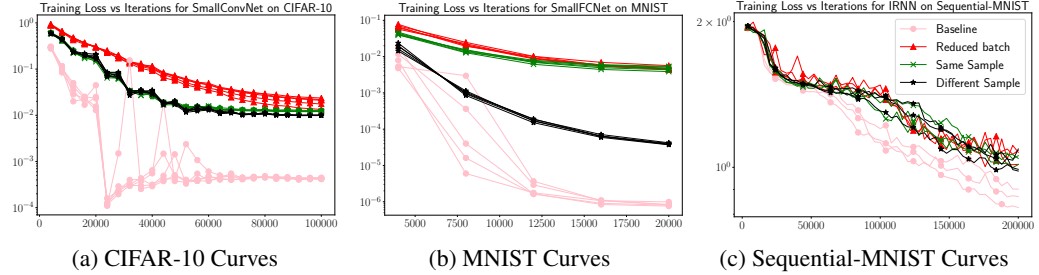

| (a) CIFAR-10 Curves | (b) MNIST Curves | (c) Sequential-MNIST Curves |

Figure 6: Training curves for neural networks. The legend in (c) applies to all plots. For the convolutional and fully connected neural networks, the loss decreases faster using activation sampling, compared to reducing the mini-batch size further to match the memory usage. For the fully connected NN on MNIST, it is important to sample different activations for each mini-batch element, since otherwise only part of the weight vector will get updated with each iteration. For the convolutional NN on CIFAR-10, this is not an issue due to weight tying. As expected, the full memory baseline converges quicker than the low memory versions. For the RNN on Sequential-MNIST, sampling different activations at each time-step matches the performance obtained by reducing the mini-batch size.

| Fraction of activations | Baseline (1.0) | 0.8 | 0.5 | 0.3 | 0.1 | 0.05 |
|---|---|---|---|---|---|---|
| ConvNet Mem | 23.08 | 19.19 | 12.37 | 7.82 | 3.28 | 2.14 |
| Fully Connected Mem | 2.69 | 2.51 | 2.21 | 2.00 | 1.80 | 1.75 |
| RNN Mem | 47.93 | 39.98 | 25.85 | 16.43 | 7.01 | 4.66 |

Table 1: Peak memory (including weights) in MBytes used during training with 150 mini-batch elements. Reported values are hand calculated and represent the expected memory usage of RAD under an efficient implementation.

For the feedforward networks we tune the learning rate and $\ell_2$ regularization parameter separately for each gradient estimator on a randomly held out validation set. We train with the best performing hyperparameters on bootstrapped versions of the full training set to measure variability in training. Details are in the appendix, including plots for train/test accuracy/loss, and a wider range of fraction of activations sampled. All feedforward models are trained with Adam.

In the RNN case, we also run baseline, "same sample", "different sample" and "reduced batch" experiments. The "reduced batch" experiment used a mini-batch size of 21, while the others used a mini-batch size of 150. The learning rate was fixed at $10^{-4}$ for all gradient estimators, found via a coarse grid search for the largest learning rate for which optimization did not diverge. Although we did not tune the learning rate separately for each estimator, we still expect that with a fixed learning rate, the lower variance estimators should perform better. When sampling, we sample different activations at each time-step. All recurrent models are trained with SGD without momentum.

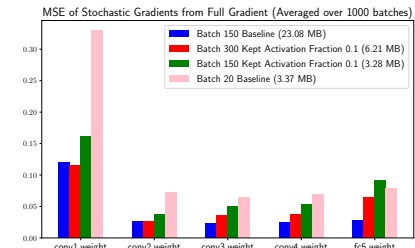

Figure 5: We visualize the gradient noise for each stochastic gradient method by computing the full gradient (over all mini-batches in the training set) and computing the mean squared error deviation for the gradient estimated by each method for each layer in the convolutional net. RAD has significantly less variance vs memory than reducing mini-batch size. Furthermore, combining RAD with an increased mini-batch size achieves similar variance to baseline 150 mini-batch elements while saving memory.

## 5 CASE STUDY: REACTION-DIFFUSION PDE-CONSTRAINED OPTIMIZATION

Our second application is motivated by the observation that many scientific computing problems involve a repeated or iterative computation resulting in a layered computational graph. We may apply RAD to get a stochastic estimate of the gradient by subsampling paths through the computational graph. For certain problems, we can leverage problem structure to develop a low-memory stochastic gradient estimator without exploding variance. To illustrate this possibility we consider the optimization of a linear reaction-diffusion PDE on a square domain with Dirichlet boundary conditions, representing the production and diffusion of neutrons in a fission reactor (McClarren,

2018). Simulating this process involves solving for a potential $\phi(x, y, t)$ varying in two spatial coordinates and in time. The solution obeys the partial differential equation:

$$\frac{\partial \phi(x, y, t)}{\partial t} = D\boldsymbol{\nabla}^2 \phi(x, y, t) + C(x, y, t, \boldsymbol{\theta})\phi(x, y, t)$$

We discretize the PDE in time and space and solve on a spatial grid using an explicit update rule $\boldsymbol{\phi}_{t+1} = \boldsymbol{M}\boldsymbol{\phi}_t + \Delta t \boldsymbol{C}_t \odot \boldsymbol{\phi}_t$, where $M$ summarizes the discretization of the PDE in space. The exact form is available in the appendix. The initial condition is $\boldsymbol{\phi}_0 = \sin(\pi x)\sin(\pi y)$, with $\phi = 0$ on the boundary of the domain. The loss function is the time-averaged squared error between $\phi$ and a time-dependent target, $L = 1/T \sum_t ||\boldsymbol{\phi}_t(\boldsymbol{\theta}) - \boldsymbol{\phi}_t^{\text{target}}||_2^2$. The target is $\boldsymbol{\phi}_t^{\text{target}} = \boldsymbol{\phi}_0 + 1/4 \sin(\pi t)\sin(2\pi x)\sin(\pi y)$. The source $C$ is given by a seven-term Fourier series in $x$ and $t$, with coefficients given by $\boldsymbol{\theta} \in \mathbb{R}^7$, where $\boldsymbol{\theta}$ is the control parameter to be optimized. Full simulation details are provided in the appendix.

The gradient is $\frac{\partial L}{\partial \boldsymbol{\theta}} = \sum_{t=1}^T \frac{\partial L}{\partial \boldsymbol{\phi}_t} \sum_{i=1}^t \left( \prod_{j=i}^{t-1} \frac{\partial \boldsymbol{\phi}_{j+1}}{\partial \boldsymbol{\phi}_j} \right) \frac{\partial \boldsymbol{\phi}_i}{\partial \boldsymbol{C}_{i-1}} \frac{\partial \boldsymbol{C}_{i-1}}{\partial \boldsymbol{\theta}}$. As the reaction-diffusion PDE is linear and explicit, $\partial \boldsymbol{\phi}_{j+1}/\partial \boldsymbol{\phi}_j \in \mathbb{R}^{N_x^2 \times N_x^2}$ is known and independent of $\phi$. We avoid storing $\boldsymbol{C}$ at each timestep by recomputing $\boldsymbol{C}$ from $\boldsymbol{\theta}$ and $t$. This permits a low-memory stochastic gradient estimate without exploding variance by sampling from $\partial L/\partial \boldsymbol{\phi}_t \in \mathbb{R}^{N_x^2}$ and the diagonal matrix $\partial \boldsymbol{\phi}_i/\partial \boldsymbol{C}_{i-1}$, replacing $\frac{\partial L}{\partial \boldsymbol{\theta}}$ with the unbiased estimator

$$\sum_{t=1}^T \mathcal{S}_{P\phi_t} \left[ \frac{\partial L}{\partial \boldsymbol{\phi}_t} \right] \sum_{i=1}^t \left( \prod_{j=i}^{t-1} \frac{\partial \boldsymbol{\phi}_{j+1}}{\partial \boldsymbol{\phi}_j} \right) \mathcal{S}_{P\phi_{i-1}} \left[ \frac{\partial \boldsymbol{\phi}_i}{\partial \boldsymbol{C}_{i-1}} \right] \frac{\partial \boldsymbol{C}_{i-1}}{\partial \boldsymbol{\theta}} . \tag{8}$$

This estimator can reduce memory by as much as 99% without harming optimization; see Figure 7b.

## 6 RELATED WORK

**Approximating gradients and matrix operations** Much thought has been given to the approximation of general gradients and Jacobians. We draw inspiration from this literature, although our main objective is designing an unbiased gradient estimator, rather than an approximation with bounded accuracy. Abdel-Khalik et al. (2008) accelerate Jacobian accumulation via random projections, in a similar manner to randomized methods for SVD and matrix multiplication. Choromanski & Sindhwani (2017) recover Jacobians in cases where AD is not available by performing a small number of function evaluations with random input perturbations and leveraging known structure of the Jacobian (such as sparsity and symmetry) via compressed sensing.

Other work aims to accelerate neural network training by approximating operations from the forward and/or backward pass. Sun et al. (2017) and Wei et al. (2017) backpropagate sparse gradients, keeping only the top $k$ elements of the adjoint vector. Adelman & Silberstein (2018) approximate matrix multiplications and convolutions in the forward pass of neural nets nets using a column-row sampling scheme similar to our subsampling scheme. Their method also reduces the computational cost of the backwards pass but changes the objective landscape.

Related are invertible and reversible transformations, which remove the need to save intermediate variables on

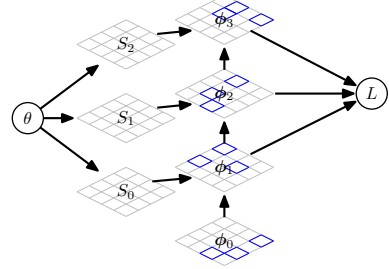

(a) Visualization of sampling

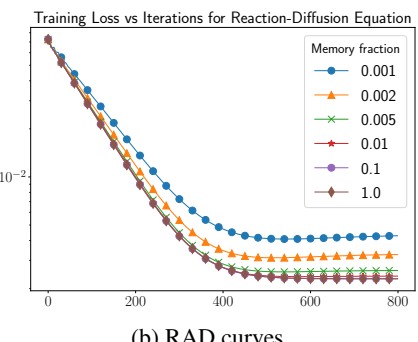

(b) RAD curves

Figure 7: Reaction-diffusion PDE expt. (b) RAD saves up to 99% of memory without significant slowdown in convergence.

the forward pass, as these can be recomputed on the backward pass. Maclaurin et al. (2015) use this idea for hyperparameter optimization, reversing the dynamics of SGD with momentum to avoid the expense of saving model parameters at each training iteration. Gomez et al. (2017) introduce a reversible ResNet (He et al., 2016) to avoid storing activations. Chen et al. (2018) introduce Neural ODEs, which also have constant memory cost as a function of depth.

**Limited-memory learning and optimization** Memory is a major bottleneck for reverse-mode AD, and much work aims to reduce its footprint. Gradient checkpointing is perhaps the most well known, and has been used for both reverse-mode AD (Griewank & Walther, 2000) with general layerwise computation graphs, and for neural networks (Chen et al., 2016). In gradient checkpointing, some subset of intermediate variables are saved during function evaluation, and these are used to re-compute downstream variables when required. Gradient checkpointing achieves sublinear memory cost with the number of layers in the computation graph, at the cost of a constant-factor increase in runtime.

**Stochastic Computation Graphs** Our work is connected to the literature on stochastic estimation of gradients of expected values, or of the expected outcome of a stochastic computation graph. The distinguishing feature of this literature (vs. the proposed RAD approach) is that it uses stochastic estimators of an objective value to derive a stochastic gradient estimator, i.e., the forward pass is randomized. Methods such as REINFORCE (Williams, 1992) optimize an expected return while avoiding enumerating the intractably large space of possible outcomes by providing an unbiased stochastic gradient estimator, i.e., by trading computation for variance. This is also true of mini-batch SGD, and methods for training generative models such as contrastive divergence (Hinton, 2002), and stochastic optimization of evidence lower bounds (Kingma & Welling, 2013). Recent approaches have taken intractable deterministic computation graphs with special structure, i.e. involving loops or the limits of a series of terms, and developed tractable, unbiased, randomized telescoping series-based estimators for the graph's output, which naturally permit tractable unbiased gradient estimation (Tallec & Ollivier, 2017; Beatson & Adams, 2019; Chen et al., 2019; Luo et al., 2020).

## 7 CONCLUSION

We present a framework for randomized automatic differentiation. Using this framework, we construct reduced-memory unbiased estimators for optimization of neural networks and a linear PDE. Future work could develop RAD formulas for new computation graphs, e.g., using randomized rounding to handle arbitrary activation functions and nonlinear transformations, integrating RAD with the adjoint method for PDEs, or exploiting problem-specific sparsity in the Jacobians of physical simulators. The randomized view on AD we introduce may be useful beyond memory savings: we hope it could be a useful tool in developing reduced-computation stochastic gradient methods or achieving tractable optimization of intractable computation graphs.

## ACKNOWLEDGEMENTS

The authors would like to thank Haochen Li for early work on this project. We would also like to thank Greg Gundersen, Ari Seff, Daniel Greenidge, and Alan Chung for helpful comments on the manuscript. This work is partially supported by NSF IIS-2007278.

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

## Appendix A: Neural Network Experiments

### Random Projections for RAD

As mentioned in Section 3.2 (around Equation 5) of the main paper, we could also use different matrices $P$ that have the properties

$$\mathbb{E}P = I_d \qquad \text{and} \qquad P = RR^T, R \in \mathbb{R}^{d \times k}, k < d\,.$$

In the appendix we report experiments of letting $R$ be a matrix of iid Rademacher random variables, scaled by $\sqrt{k}$. $P = RR^T$ defined in this way satisfies the properties above. Note that this would lead to additional computation: The Jacobian or input vector would have to be fully computed, and then multiplied by $R$ and stored. In the backward pass, it would have to be multiplied by $R^T$. We report results as the "project" experiment in the full training/test curves in the following sections. We see that it performs competitively with reducing the mini-batch size.

### Architectures Used

We use three different neural network architectures for our experiments: one fully connected feedforward, one convolutional feedforward, and one recurrent.

Our fully-connected architecture consists of:

1. Input: 784-dimensional flattened MNIST Image
2. Linear layer with 300 neurons (+ bias) (+ ReLU)
3. Linear layer with 300 neurons (+ bias) (+ ReLU)
4. Linear layer with 300 neurons (+ bias) (+ ReLU)
5. Linear layer with 10 neurons (+ bias) (+ softmax)

Our convolutional architecture consists of:

1. Input: $3 \times 32 \times 32$-dimensional CIFAR-10 Image
2. $5 \times 5$ convolutional layer with 16 feature maps (+ 2 zero-padding) (+ bias) (+ ReLU)
3. $5 \times 5$ convolutional layer with 32 feature maps (+ 2 zero-padding) (+ bias) (+ ReLU)
4. $2 \times 2$ average pool 2-d
5. $5 \times 5$ convolutional layer with 32 feature maps (+ 2 zero-padding) (+ bias) (+ ReLU)
6. $5 \times 5$ convolutional layer with 32 feature maps (+ 2 zero-padding) (+ bias) (+ ReLU)
7. $2 \times 2$ average pool 2-d (+ flatten)
8. Linear layer with 10 neurons (+ bias) (+ softmax)

Our recurrent architecture was taken from Le et al. (2015) and consists of:

1. Input: A sequence of length 784 of 1-dimensional pixels values of a flattened MNIST image.
2. A single RNN cell of the form

$$h_t = \text{ReLU}(W_{ih}x_t + b_{ih} + W_{hh}h_{t-1} + b_{hh})$$

   where the hidden state ($h_t$) dimension is 100 and $x_t$ is the 1-dimensional input.
3. An output linear layer with 10 neurons (+ bias) (+ softmax) that has as input the last hidden state.

### Calculation of memory saved from RAD

For the baseline models, we assume inputs to the linear layers and convolutional layers are stored in 32-bits per dimensions. The ReLU derivatives are then recalculated on the backward pass.

For the RAD models, we assume inputs are sampled or projected to 0.1 of their size (rounded up) and stored in 32-bits per dimension. Since ReLU derivatives can not exactly be calculated now, we

assume they take 1-bit per dimension (non-reduced dimension) to store. The input to the softmax layer is not sampled or projected.

In both cases, the average pool and bias gradients does not require saving since the gradient is constant.

For MNIST fully connected, this gives (per mini-batch element memory):

Baseline: $(784 + 300 + 300 + 300 + 10) \cdot 32$ bits $= 6.776$ kBytes

RAD 0.1: $(79 + 30 + 30 + 30 + 10) \cdot 32$ bits $+ (300 + 300 + 300) \cdot 1$ bits $= 828.5$ bytes

which leads to approximately 8x savings per mini-batch element.

For CIFAR-10 convolutional, this gives (per mini-batch element memory):

Baseline: $(3 \cdot 32 \cdot 32 + 16 \cdot 32 \cdot 32 + 32 \cdot 16 \cdot 16 + 32 \cdot 16 \cdot 16 + 32 \cdot 8 \cdot 8 + 10) \cdot 32$ bits $= 151.59$ kBytes

RAD 0.1: $(308 + 1639 + 820 + 820 + 205 + 10) \cdot 32$ bits $+ (16384 + 8192 + 8192 + 2048) \cdot 1$ bits $= 19.56$ kBytes

which leads to approximately 7.5x savings per mini-batch element.

For Sequential-MNIST RNN, this gives (per mini-batch element memory):

Baseline: $(784 \cdot (1 + 100) + 100 + 10) \cdot 32$ bits $= 317.176$ kBytes

RAD 0.1: $(784 \cdot (1 + 10) + 10 + 10) \cdot 32$ bits $+ (784 \cdot 100) \cdot 1$ bits $= 44.376$ kBytes

which leads to approximately 7.15x savings per mini-batch element.

### FEEDFORWARD NETWORK TRAINING DETAILS

We trained the CIFAR-10 models for $100,000$ gradient descent iterations with a fixed mini-batch size, sampled with replacement from the training set. We lower the learning rate by $0.6$ every $10,000$ iterations. We train with the Adam optimizer. We center the images but do not use data augmentation. The MNIST models were trained similarly, but for $20,000$ iterations, with the learning rate lowered by $0.6$ every $2,000$ iterations. We fixed these hyperparameters in the beginning and did not modify them.

We tune the initial learning rate and $\ell_2$ weight decay parameters for each experiment reported in the main text for the feedforward networks. For each experiment (project, same sample, different sample, baseline, reduced batch), for both architectures, we generate 20 (weight decay, learning rate) pairs, where each weight decay is from the loguniform distribution over $0.0000001 - 0.001$ and learning rate from loguniform distribution over $0.00001 - 0.01$.

We then randomly hold out a validation dataset of size 5000 from the CIFAR-10 and MNIST training sets and train each pair on the reduced training dataset and evaluate on the validation set. For each experiment, we select the hyperparameters that give the highest test accuracy.

For each experiment, we train each experiment with the best hyperparameters 5 times on separate bootstrapped resamplings of the full training dataset ($50,000$ for CIFAR-10 and $60,000$ for MNIST), and evaluate on the test dataset ($10,000$ for both). This is to make sure the differences we observe across experiments are not due to variability in training. In the main text we show 3 randomly selected training curves for each experiment. Below we show all 5.

All experiments were run on a single NVIDIA K80 or V100 GPU. Training times were reported on a V100.

### RNN TRAINING DETAILS

All RNN experiments were trained for 200,000 iterations (mini-batch updates) with a fixed mini-batch size, sampled with replacement from the training set. We used the full MNIST training set of 60,000 images whereby the images were centered. Three repetitions of the same experiment were performed with different seeds. Hyperparameter tuning was not performed due to time constraints.

The hidden-to-hidden matrix ($W_{hh}$) is initialised with the identity matrix, the input-to-hidden matrix ($W_{ih}$) and hidden-to-output (last hidden layer to softmax input) are initialised with a random matrix where each element is drawn independently from a $\mathcal{N}(0, 0.001)$ distribution and the biases ($b_{ih}$, $b_{hh}$) are initialised with zero.

The model was evaluated on the test set of 10,000 images every 400 iterations and on the entire training set every 4000 iterations.

For the "sample", "different sample", "project" and "different project" experiments different activations/random matrices were sampled at every time-step of the unrolled RNN.

All experiments were run on a single NVIDIA K80 or V100 GPU.

The average running times for each experiment are given in Table 2. Note that we did not optimise our implementation for speed and so these running times can be reduced significantly.

Table 2: Average running times for RNN experiments on Sequential-MNIST.

| Experiment | Running Time (hrs) | GPU |
|---|---|---|
| Baseline | 34.0 | V100 |
| Small batch | 16.0 | V100 |
| Sample | 96.0 | K80 |
| Different Sample | 110.0 | K80 |
| Project | 82.0 | K80 |
| Different Project | 89.0 | K80 |

IMPLEMENTATION

The code is provided on GitHub[1]. Note that we did not optimize the code for computational efficiency; we only implemented our method as to demonstrate the effect it has on the number of gradient steps to train. Similarly, we did not implement all of the memory optimizations that we account for in our memory calculations; in particular in our implementation we did not take advantage of storing ReLU derivatives with 1-bit or the fact that average pooling has a constant derivative. Although these would have to be implemented in a practical use-case, they are not necessary in this proof of concept.

---

[1]https://github.com/PrincetonLIPS/RandomizedAutomaticDifferentiation

FULL TRAINING/TEST CURVES FOR MNIST AND CIFAR-10

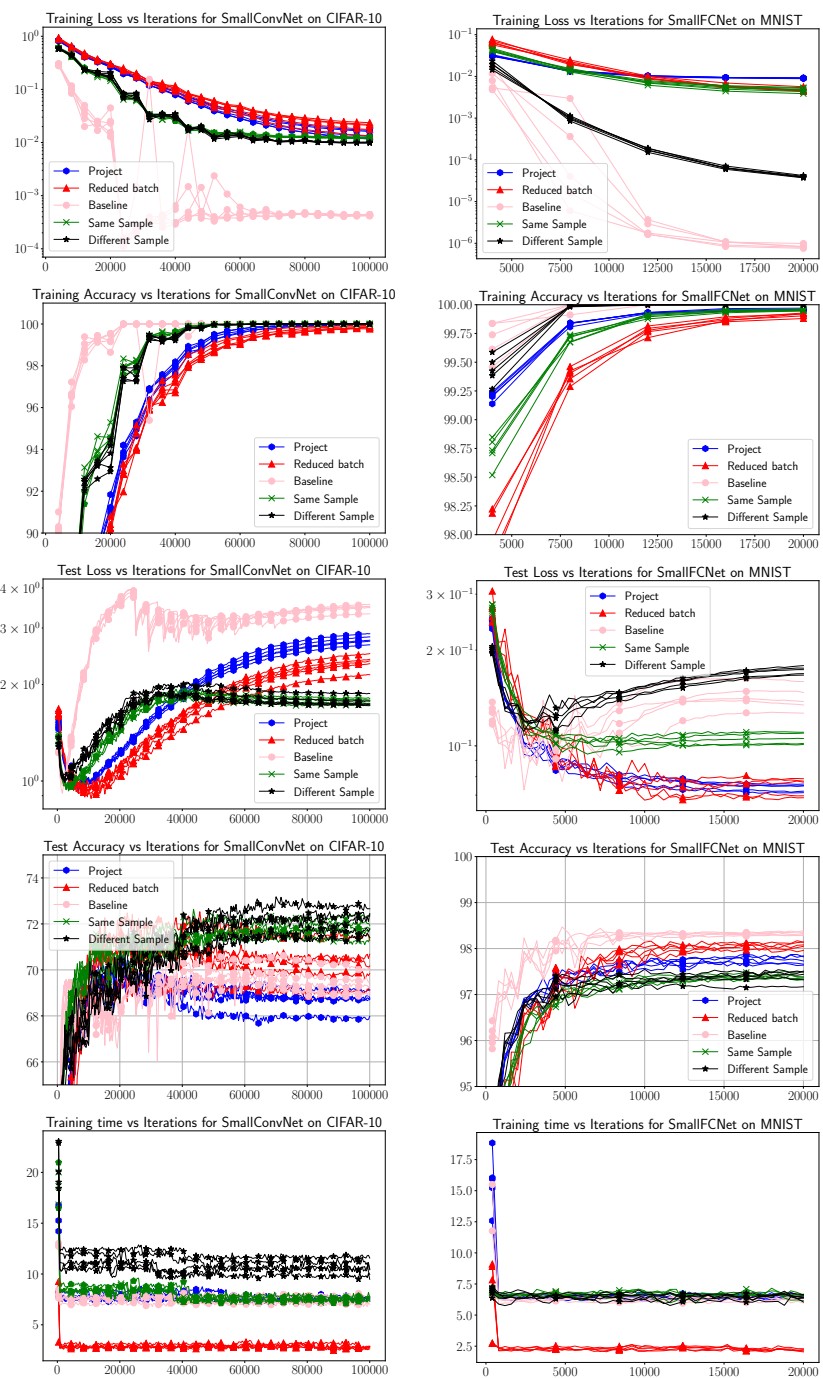

Figure 8: Full train/test curves and runtime per iteration. Note that training time is heavily dependent on implementation, which we did not optimize. In terms of FLOPs, "project" should be significantly higher than all the others and "reduced batch" should be significantly smaller. The "baseline", "same sample", and "different sample" should theoretically have approximately the same number of FLOPs.

FULL TRAINING/TEST CURVES FOR RNN ON SEQUENTIAL-MNIST

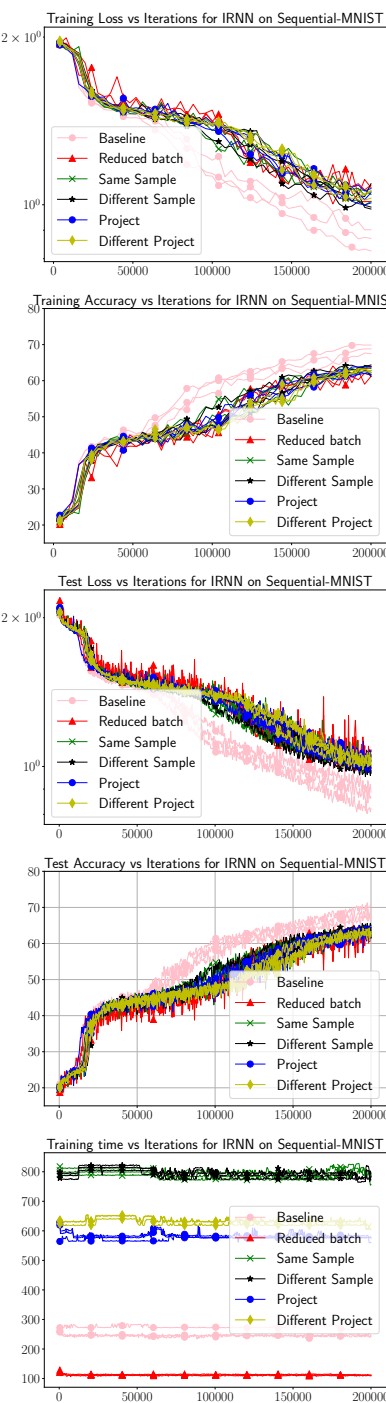

Figure 9: Full train/test curves and runtime per 400 iterations. We also include results for random projections with shared and different random matrices for each mini-batch element.

Sᴡᴇᴇᴘ ᴏꜰ ꜰʀᴀᴄᴛɪᴏɴs sᴀᴍᴘʟᴇᴅ ꜰᴏʀ MNIST ᴀɴᴅ CIFAR-10

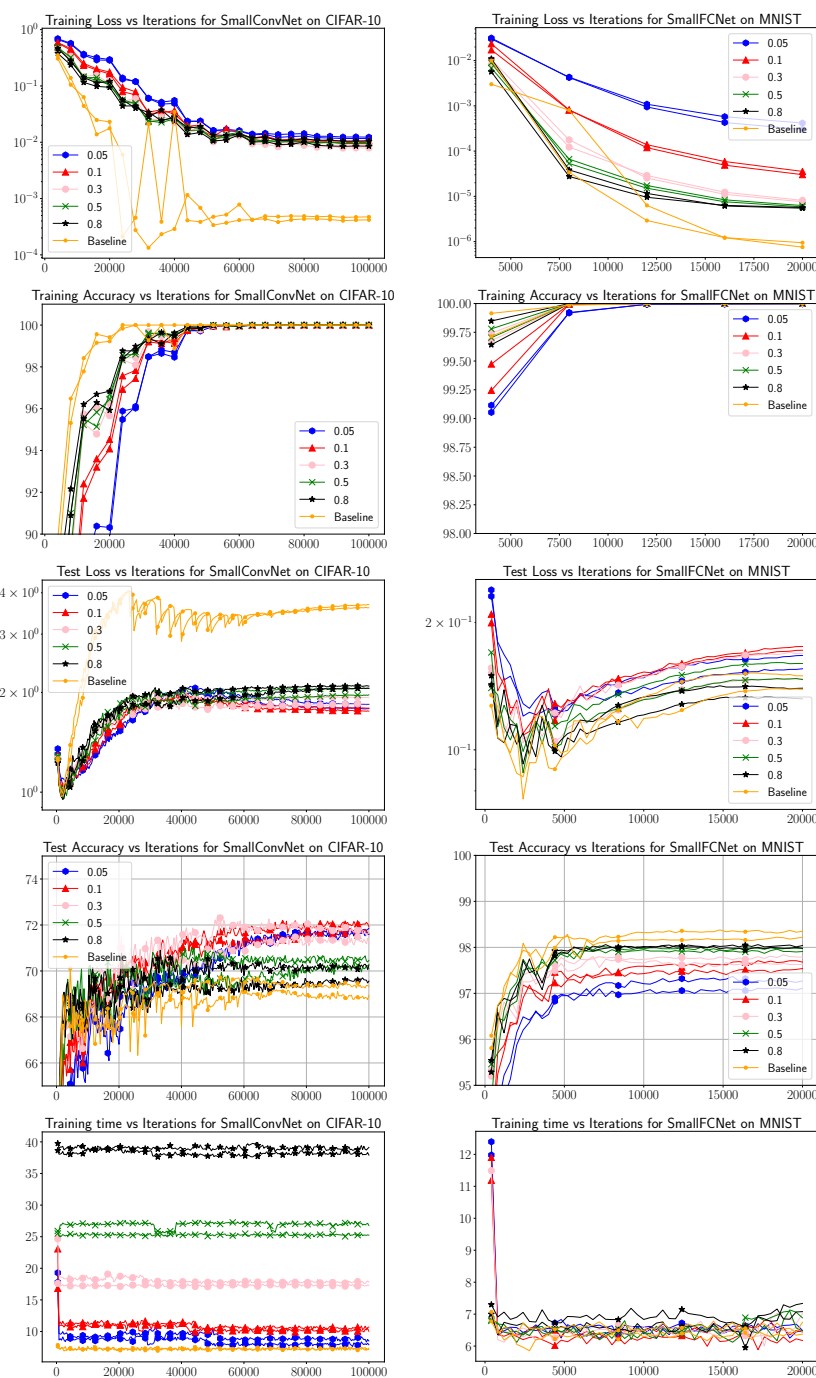

Figure 10: Full train/test curves and runtime per iteration for various fractions for the "different sample" experiment. Note that the reason 0.8 does not quite converge to baseline in the training curve is because we sample with replacement. This is an implementation detail; our method could be modified to sample without replacement, and at fraction 1.0 would be equivalent to baseline. The weight decay and initial learning rate for the RAD experiments above are all the same as the ones tuned for 0.1 fraction "different sample" experiment. The baseline experiments are tuned for baseline.

## APPENDIX B: REACTION-DIFFUSION PDE

The reaction-diffusion equation is a linear parabolic partial differential equation. In fission reactor analysis, it is called the one-group diffusion equation or one-speed diffusion equation, shown below.

$$\frac{\partial \phi}{\partial t} = D\boldsymbol{\nabla}^2 \phi + C\phi + S$$

Here $\phi$ represents the neutron flux, $D$ is a diffusion coefficient, and $C\phi$ and $S$ are source terms related to the local production or removal of neutron flux. In this paper, we solve the one-speed diffusion equation in two spatial dimensions on the unit square with the condition that $\phi = 0$ on the boundary. We assume that $D$ is constant equal to $1/4$, $C(x, y, t, \boldsymbol{\theta})$ is a function of control parameters $\boldsymbol{\theta}$ described below, and $S$ is zero. We discretize $\phi$ on a regular grid in space and time, which motivates the notation $\phi \to \phi_t$. The grid spacing is $\Delta x = 1/32$ and the timestep is $\Delta t = 1/4096$. We simulate from $t = 0$ to $t = 10$. We use the explicit forward-time, centered-space (FTCS) method to timestep $\phi$. The timestep is chosen to satisfy the stability criterion, $D\Delta t/(\Delta x)^2 \leq \frac{1}{4}$. In matrix notation, the FTCS update rule can be written $\phi_{t+1} = \boldsymbol{M}\phi_t + \Delta t \boldsymbol{C}_t \odot \phi_t$, in index notation it can be written as follows:

$$\phi_{t+1}^{i,j} = \phi_t^{i,j} + \frac{D\Delta t}{(\Delta x)^2}\left(\phi_t^{i+1,j} + \phi_t^{i-1,j} + \phi_t^{i,j+1} + \phi_t^{i,j-1} - 4\phi_t^{i,j}\right) + \Delta t C_t^{i,j} \phi_t^{i,j}$$

The term $C\phi$ in the one-speed diffusion equation relates to the local production or removal of neutrons due to nuclear interactions. In a real fission reactor, $C$ is a complicated function of the material properties of the reactor and the heights of the control rods. We make the simplifying assumption that $C$ can be described by a 7-term Fourier series in $x$ and $t$, written below. Physically, this is equivalent to the assumption that the material properties of the reactor are constant in space and time, and the heights of the control rods are sinusoidally varied in $x$ and $t$. $\phi_0$ is initialized so that the reactor begins in a stable state, the other parameters are initialized from a uniform distribution.

$$C(x, y, t, \boldsymbol{\theta}) = \theta_0 + \theta_1 \sin(\pi t) + \theta_2 \cos(\pi t) + \theta_3 \sin(2\pi x)\sin(\pi t)+$$
$$\theta_4 \sin(2\pi x)\cos(\pi t) + \theta_5 \cos(2\pi x)\sin(\pi t) + \theta_6 \cos(2\pi x)\cos(\pi t)$$

The details of the stochastic gradient estimate and optimization are described in the main text. The Adam optimizer is used. Each experiment of 800 optimization iterations runs in about 4 hours on a GPU.

APPENDIX C: PATH SAMPLING ALGORITHM AND ANALYSIS

Here we present an algorithm for path sampling and provide a proof that it leads to an unbiased estimate for the gradient. The main idea is to sample edges from the set of outgoing edges for each vertex in topological order, and scale appropriately. Vertices that have no incoming edges sampled can be skipped.

---

**Algorithm 1** RMAD with path sampling

---

 1: **Inputs**:
 2:     $\mathcal{G} = (V, E)$ - Computational Graph. $d_v$ denotes outdegree, $v.succ$ successor set of vertex $v$.
 3:     $y$ - Output vertex
 4:     $\Theta = (\theta_1, \theta_2, \ldots, \theta_m) \subset V$ - Input vertices
 5:     $k > 0$ - Number of samples per vertex
 6: **Initialization**:
 7:     $Q(e) = 0, \forall e \in E$
 8: **for** $v$ in topological order; synchronous with forward computation **do**
 9:     **if** No incoming edge of $v$ has been sampled **then**
10:         Continue
11:     **for** $k$ times **do**
12:         Sample $i$ from $[d_v]$ uniformly.
13:         $Q(v, v.succ[i]) \leftarrow Q(v, v.succ[i]) + \frac{d_v}{k}\frac{\partial v.succ[i]}{\partial v}$
14: Run backpropagation from $y$ to $\Theta$ using $Q$ as intermediate partials.
15: **Output**: $\nabla_\Theta y$

---

The main storage savings from Algorithm 1 will come from Line 9, where we only consider a vertex if it has an incoming edge that has been sampled. In computational graphs with a large number of independent paths, this will significantly reduce memory required, whether we record intermediate variables and recompute the LCG, or store entries of the LCG directly.

To see that path sampling gives an unbiased estimate, we use induction on the vertices in reverse topological order. For every vertex $z$, we denote $\bar{z} = \frac{\partial y}{\partial z}$ and $\hat{z}$ as our approximation for $\bar{z}$. For our base case, we let $\hat{y} = \frac{dy}{dy} = 1$, so $\mathbb{E}\hat{y} = \bar{y}$. For all other vertices $z$, we define

$$\hat{z} = d_z \sum_{(z,v)\in E} \mathbb{I}_{v=v_i}\frac{\partial v}{\partial z}\hat{v} \tag{9}$$

where $d_z$ is the out-degree of $z$, $v_i$ is sampled uniformly from the set of successors of $z$, and $\mathbb{I}_{v=v_i}$ is an indicator random variable denoting if $v = v_i$. We then have

$$\mathbb{E}\hat{z} = \sum_{(z,v)\in E} d_z\mathbb{E}[\mathbb{I}_{v=v_i}]\frac{\partial v}{\partial z}\mathbb{E}\hat{v} = \sum_{(z,v)\in E} \frac{\partial v}{\partial z}\mathbb{E}\hat{v} \tag{10}$$

assuming that the randomness over the sampling of outgoing edges is independent of $\hat{v}$, which must be true because our induction is in reverse topological order. Since by induction we assumed $\mathbb{E}\hat{v} = \bar{v}$, we have

$$\mathbb{E}\hat{z} = \sum_{(z,v)\in E} \frac{\partial v}{\partial z}\bar{v} = \bar{z} \tag{11}$$

which completes the proof.

