# OpenReview forum: "Randomized Automatic Differentiation"
_ICLR.cc/2021/Conference — ICLR 2021 Oral_

### Official Review · AnonReviewer4 · 2020-10-25
**Technical explanation is not sufficient.**

**Rating:** 7
**Confidence:** 4

**Review:**

The paper proposes a novel approach to reduce memory in backpropagation by sampling the paths in DAG. The paper developed and proved the proposed method in a general framework (which is nice!). Still, I feel that the explanation for applying this method to neural networks is somewhat lacking. I summarize the missing parts in the following.

(1) For fully-connected layers (matrix multiplication), the explanations in Section 3.2 and Section 4.2 are not consistent --- Section 3.2 suggests sampling from the paths (connections), while Section 4.2 suggests sampling from the vertices (activations). I believe in practice sampling from activations is used (correct me if I am mistaken), which indeed implies that random matrices {P_1, ..., P_n} are not independent (I guess it forms a Markov chain?). Therefore, the analysis in Section 3.2 is not applicable. I think an analysis using dependent sampling (ancestral sampling) is needed in this case.

(2) The explanation for convolutional layers is not sufficient. The paper mentions that the compact structure could be exploited multiple times, but exactly how it is somewhat missing. I am particularly wondering whether the activations (pixels) or the feature maps (channels) are sampled. If the activations are sampled, I think it is not friendly to the CUDA kernels as the activations within the same channel are computed together; And if the feature maps are sampled, it again violates the analysis in Section 3.2 (since some paths must be chosen together as a group).

(3) It is not clear how the proposed method is compatible with normalization layers. In normalization layers (e.g., batch normalization), the statistics of the activations are computed. However, since the activations now are sampled, it is not explained how the normalization layers should be modified accordingly to preserve unbiasedness.

(4) It is not clear how the proposed method is compatible with randomized operations in neural networks (e.g., dropout, reparameterization, etc.) I am wondering if the combination is still unbiased.

Given that the technical explanation is lacking for the moment, it is hard for me to judge this paper's correctness. I will temporarily give a reject. I am very willing to increase my score if the authors address the confusion above.

---

> ### Author Response · Authors · 2020-11-16
> **Response to reviewer #4**
>
> Thank you for taking the time to review our paper! We would like to clear up the technical explanation:
>
> 1. In section 3.2, the interpretation of right multiplying dC/dB (see equation 4) by a random matrix is to sample vertices from B and only keep paths going through those vertices. The random matrix, with proper weighting has expectation identity. Furthermore, the calculation of each gradient (for a weight) only involves multiplying by a single random matrix (it would be multiplied on the right of the last term in the matrix chain to calculate the gradient, first equation on page 6). Therefore, even if P_1, P_2 were correlated for different activations, they would not affect each other as each gradient term involves a single P. We will clear this up in the text. Thank you for pointing out this possible confusion!
>
> 2. For convolutional networks we do indeed sample activations (pixels). Our goal here though is to save memory, not computation. In our implementation (in the appendix), we store a compact version of the activations during the forward pass, and reconstruct a sparse version of the activations in the backward pass. As you mentioned, CUDA kernels would compute the activations together so this would certainly not save compute, but it will save the peak memory required during training.
>
> 3. Unfortunately the current version of the method is not designed to work with normalization layers, although this is a central focus for future work.
>
> 4. The combination of RAD with randomized operations such as dropout and the reparameterization trick will be unbiased. Consider a neural network. We know we can apply RAD to this neural network to get an unbiased low-memory gradient estimate. Now consider that neural network with dropout. For a given data point and dropout mask, this is equivalent to another neural network with some of the weights set to zero. RAD can be applied to get a low-memory unbiased gradient estimate for this second neural network. RAD thus provides an unbiased estimate of the objective for any given specific dropout mask, which itself an unbiased estimate of the expected loss across all dropout masks. Therefore, RAD can be composed with dropout to get an unbiased gradient estimate (of the training objective of the stochastic model that is the NN with dropout). A similar argument can be applied to reparameterization trick models by considering these just as NNs with an extra input (which is the random vector drawn from e.g. a Gaussian or uniform distribution which is used to generate the reparameterized random variables).

---

> > ### Comment · AnonReviewer4 · 2020-11-17
> > **Thanks for your clarification.**
> >
> > Thanks for the technical explanation.  The clarifications convince me from a high level, and I will update my scores after the manuscript update.
> >
> > However, I have a follow-up question: Since the random matrices are correlated anyway, is there a principled way to derive the sampling strategy with the lowest variance?  The question is motivated by your explanation that the pixels are sampled instead of feature maps. I am wondering about the principle behind this choice.

---

> > > ### Author Response · Authors · 2020-11-17
> > > **Correlated P_1, P_2**
> > >
> > > Thanks for the followup.
> > >
> > > We would like to clarify. There may have been some miscommunication about which random matrices we are talking about.
> > >
> > > In page 4 of the manuscript: {P_1, P_2, P_3, ..., P_i} are matrices that are composed as outer products of the ith basis vectors. These are not random matrices. To create random matrices, we draw from these uniformly with replacement, and take their scaled average (matrix P defined in the text near the top of page 4).
> > >
> > > For each activation, we independently do this sampling, so that the P matrices corresponding to each activation (say, P_A, P_B) are mutually independent. We interpret this as "sampling activations" because when we right multiply the Jacobian dC/dB by P it is equivalent to sampling vertices from B. The unbiasedness of the full gradient follows directly from equation 5 (in the now updated manuscript).
> > >
> > > In practice we do not need to generate the full random matrices. All we need to do is draw a binary mask with which to multiply B (thereby "sampling vertices") and scale the result appropriately: we then do not need to store the masked-out components of B for reverse-mode AD.
> > >
> > > The reason we sample pixels vs feature maps is because in the convolutional networks we used there were many more pixels than feature maps. There might be many other interesting sampling strategies that are interesting to explore for future work, including simple mixed strategies where in some layers we sample activations while in others we sample feature maps.
> > >
> > > In terms of variance, since the computation of each weight gradient only involves a single random matrix, then it is not possible to reduce the variance by developing a correlated sampling scheme (e.g. antithetic sampling) between the matrices. It would be possible to reduce variance by e.g. importance sampling, with importance weights on sampling vertices from B determined by the magnitude of dL/dB, but this would require you to know and store the gradient before sampling. Maybe cheap heuristics (such as the magnitude of the activations) might help, but that's beyond the scope of this paper.

---

### Official Review · AnonReviewer3 · 2020-10-25
**Well-written and solid study on the fundemental**

**Rating:** 8
**Confidence:** 4

**Review:**

In the context of deep learning, back-propagation is stochastic in the sample level to attain bette efficiency than full-dataset gradient descent. The authors asked that, can we further randomize the gradient compute within each single minibatch / sample with the goal to achieve strong model accuracy. In modern deep learning, training memory consumption is high due to activation caching. Thus this randomized approach can help attain strong model accuracy under memory constraints.

The authors proposed a general framework for randomized auto differentiation to achieve unbiased gradient estimators. This general framework allows randomization at different granularity such as layer level and individual neuron level. It also includes conventional minibatch gradient estimators as a special case at the sample/minibatch level for randomization. The memory saving here is achieved by trading off gradient variance for activation memory saving.

Empirically, the authors show that for 1) convolution nets on MNIST and CIFAR and 2) RNN on sequential-MNIST, under the same memory budget, neuron-level randomized gradient estimator can achieve higher model accuracy than conventional SGD with smaller minibatch size.

Strong point: This paper is well written with novel thoughts on the fundamental aspects of auto-diff when applied to deep learning (and also to PDE as demoed in section 5.). It can also provide new options for practitioners to train models with high accuracy under memory constraints. Thus I recommend to accept this paper.

I have the following comments / questions on the technical aspects. I only raise these questions up for improvement on the paper; they are not concerns on the quality of the current version. Nonetheless,  I am happy to raise the score if the authors can demonstrate results on these aspects.

1. The authors mentioned about leveraging model specific structures to control/reduce gradient variance for fine-grained randomization (such as at the neuron level). Specifically, they considered using the randomized activation sparsification only for activation memory-consuming layers. I was wondering if other model structures can also help here. E.g. can we sample at the full-channel level for convolutional layers or column / row level for linear layers. Would this has a significant impact of the attained model accuracy?

2. The main focus on practical implications in this paper is about high accuracy training under memory constraints. However, I was wondering how the authors think about the implication on compute, especially related to the sparse training trend. E.g. can we also make the forward compute itself also sparse and still minimally influence model accuracy?


NITs:

1. First letter capitalization for figure 3 at the beginning of section 4.

2. Would it be possible to provide some preliminary in appendix on the solution to the section 5 PDE example.

3. In section 3.3, the discussion on two extremal cases needs a more precision in text. Assuming the sampling on each connection (i.e. segment on paths) are independent, I think both case will have variance exponential in terms of depth. Currently it reads like only the fully connected case have exponentially large variance.

---

> ### Author Response · Authors · 2020-11-16
> **Response to reviewer #3**
>
> We are glad you enjoyed the paper! We would first like to clarify one thing from your review:
>
> - The goal of our methods is not to optimize for model accuracy. We understand that generalization in neural networks is highly brittle and depends on many hyperparameters and architecture-specific details. We therefore decided to compare the methods solely based on the number of gradient iterations needed to decrease the loss, which is what is reported in Figure 5 (now 6 in the updated paper). This is one of the reasons why we focused on very simple models with very simple training schemes: We wanted a controlled experiment where the details are as simple as possible. In the appendix, we give all the results, including final test/train accuracy (see e.g. Figure 8).
>
> Using model structures:
>
> - This is something that could be worth investigating. We currently only sampled feature maps (pixels) in the case of CNNs and neurons (activations) in the case of fully connected nets, but certainly many structured sampling schemes could be possible.
>
> Optimizing compute in the forward pass:
>
> - Sparse model training is indeed a very interesting area! It is quite possible that some of the ideas from RAD could be useful there. But there is a fundamental difference, at least for the purpose of this paper: Sparsifying the forward pass changes the function that the model is computing. The gradient computed with a sparse forward pass is a biased estimate of the gradient computed with the full model, and RAD cannot be used directly to construct a sparse forward pass unbiased estimator. In this paper, we focussed on keeping the forward pass identical while modifying the backward pass to get an unbiased gradient.
> - Computing unbiased gradients from a sparse forward pass would be extremely interesting, but it’s unclear whether this is possible (at least for a fixed sparsity level), and if possible would require some different methodology.
>
> NITs: Thanks for pointing these out. We will clarify. In particular, in the section 3.3 extremal cases we mean to sample paths rather than connections. In this case, the left example would not be exponential, because there will be a constant number of paths with respect to depth.

---

### Official Review · AnonReviewer2 · 2020-10-27
**Original idea with a lot of potential**

**Rating:** 8
**Confidence:** 4

**Review:**

The authors introduce the novel idea of producing unbiased gradient estimates by Monte Carlo sampling the paths in the autodiff linearized computational graph.
Based on sampling paths it is possible to save memory due to not having to store the complete linearized computational graph (or the intermediate variables necessary to reconstruct it).
Memory is the main bottleneck for reverse mode autodiff for functions with lots of floating point operations (such as a numerical integrator that performs many time steps, or a very deep neural network).
The authors' idea can therefore potentially enable the gradient based optimization of objective functions with many floating point operations without check pointing and recomputing to reverse through the linearized computational graph.
The tradeoff made is the introduction of (additional) variance in the gradient estimates.

The basic idea is simple and elegant:
The linearized computational graph of a numerical algorithm is obtained by
a) having the intermediate variables of the program as vertices
b) drawing directed edges from the right-hand side variables of an assignment to its left-hand side variable
c) labeling the edges by the (local) partial derivatives of assignments' left-hand side with respect to their right-hand side.

The derivative of an "output" y with respect to an "input" x of the function is the sum over all paths from x to y through the linearized computational graph taking the product of all the edges in the path.
The sum over all paths corresponds to the expectation of a uniform distribution over the paths times the number of paths.
That expectation can be Monte Carlo sampled.

The authors suggest a way of producing the path sampling based on taking a chained matrix view of the computation graph (see e.g https://arxiv.org/abs/2003.05755) and injecting low rank random matrices.
Due to the fact that the expectation of the product of independent random variables is equal to the product of the expectations this injection is unbiased as well if the injected matrices have the identity matrix as expectation.

To take advantage of the simple idea it is in practice necessary to consider the concrete shape of the computational graph at hand in order to decide where to best randomize and save memory without letting variance increase too much.

The authors present a neural network case study where they show that for some architectures the suggested approach has a better memory to variance trade off than simply choosing a smaller mini-batch size.
Furthermore, they present a 2D PDE solver case study where their approach can save a lot of memory and still optimize well.

I recommend to accept the paper.

Remarks:

I would love to see a more in depth analysis of the variance for example for simple but insightful toy examples.

For exampl simple sketching with random variates v with E[vv^T] = I can be used to obtain an unbiased gradient estimate via E[gvv^T] = g, i.e. by evaluating a single forward-mode AD pass (or just a finite difference perturbation).
But of course the gradient estimate has such a high variance so as to not give any advantage over finite difference methods (since with each sample / evaluation we are only capturing one direction in the input space).
We are not gaining the usual benefit of reverse mode autodiff of getting information about the change of the function in all directions.

In order for paths to be efficiently Monte Carlo-friendly it is probably necessary that they are correlated with other paths.
In practice this will perhaps have something to do with e.g. the regularity of the PDE solution (the gradient with respect to the solution is similar to that of its neighborhood).

A simple example (ODE integrator):

p1 = x
p2 = x
for i in range(n):
	p1 = p1 + h * sin(p1)
	p2 = p2 + h * sin(p2)

y = 0.5 * (p1 + p2)

The two paths in the program compute exactly the same values so leaving one path out randomly does not make any difference at all (if we correctly re-weight the estimate).

Mini-batches are often like that: Independent samples from the same class give correlated computations, hence the variance is related to the variance in the data.

But if two paths involve completely independent and uncorrelated computations then the variance is such that we do not gain anything.
We need at least two gradient steps to incorporate the information from both paths.
Since we do not systematically cycle through them but sample randomly, we are actually going to be less efficient.

In terms of arguing about memory savings for machine learning applications it would be interesting to see a case study with a large scale architecture that does not fit into memory.

The random matrix injection section could be clarified by moving the sentence "the expectation of a product of independent random variables is the product of their expectation" further to the front and state clearly the idea that:
E[A PP^T B QQ^T C] = A E[PP^T] B E[QQ^T] C = A I B I C = A B C

In the PDE example you could clarify the notation used to properly distinguish between the continuous and the discretized solution.

Also the PDE constrained optimization problem is not inherently stochastic (as can be argued for the empirical risk minimization setting in machine learning).
Therefore, it is possible to use non-SGD methods with linear or even super-linear convergence rates (quasi-Newton methods).
SGD with unbiased gradients has a sublinear rate of convergence.
But the ideas of the paper are of course still useful even when trying to find the optimum up to machine precision in finite time.
We can first use the RAD SGD approach in the early optimization and then go to the deterministic setting later in the optimization.

- Page 3: eqn equation 1 -> Equation 1
- Page 6: figure 5 -> Figure 5
- Throughout: Perhaps clarify the meaning of batch vs mini-batch (in other papers batch can refer to full-batch)
- Figure 5 (a) has blue curves but blue is not in the legend of Figure 5 (c)
- Page 8: backpropogate -> backpropagate

---

> ### Author Response · Authors · 2020-11-16
> **Response to reviewer #2**
>
> Thank you for your very detailed review! We are glad you enjoyed the paper.
>
> Toy examples for variance:
> - We agree it would help to add some toy examples for variance in the paper. We will look into this for the final version. We have done empirical analysis on the variance, which was originally in Appendix A but have now moved to the main paper (Figure 5).
> - Thank you for your insights on this. The ODE example is good. We agree that to be Monte Carlo friendly several paths must be correlated. We will investigate this further!
>
> Large scale architectures:
> - Unfortunately a limitation for our current approach is that it does not support batch normalization, so we would have some difficulty trying it on large-scale models (although, there are ways to train large-scale models without batch normalization, such as FixUp Initialization, which we are thinking about). Getting our method to work on batch normalization is certainly a next step we are thinking about, but for the purposes of this paper we wanted to try out our ideas on simple architectures to run a controlled experiment of RAD without too many complicating details.
>
> Clarifying random matrix injection section:
> - Good call! We have updated the paper as such.
>
> PDE section:
> - We have added some clarification on the discretization, but definitely plan on making another pass through for the final version.
> - Good point on the non-stochasticity of PDEs. This is a valuable insight.
>
> Extra:
> - Thanks for catching these! Especially good call on the blue curves in the graph.

---

### Official Review · AnonReviewer1 · 2020-10-28
**Interesting work on unbiased gradient estimation with lower memory cost**

**Rating:** 7
**Confidence:** 4

**Review:**

Summary:

The paper proposes to subsample the computational graph to obtain an unbiased gradient estimator with less memory requirement, in the same spirit as minibatching reducing the memory of the full-batch gradient descent. The authors propose to do so by modifying the forward pass of a linear layer to only save a subset of the hidden units to estimate the gradient.


Contributions:

Just like minibatching is introduced to reduce the memory requirement of full-batch gradient descent, this paper introduces another dimension to trade off between noisy gradient estimate and memory by subsampling the computational graph for computing a noisy gradient. The experiments, albeit small-scale, clearly demonstrate the effect of using SGD with randomized AD to train standard neural networks. The application on stochastic optimization of PDE is also very interesting and relevant. The paper is very well written and explains the idea very clearly. I look forward to seeing future work using this principle to train larger scale models as well as how this work will enable the development of many more research ideas the same way minibatch SGD acts as a workhorse of deep learning / machine learning.


Additional details:

In the exposition of the “alternative SGD scheme with randomized AD“, it might help to understand the sources of the variance of the gradient estimator by decomposing it as Var(x_ij) = Var(x_i) + E[Var(x_ij | x_i)], where i is the index for minibatching and j is for subgraph sampling. It will also help to understand the different curves of Fig 5.

Is there any speculation of why the drop in performance is more drastic in MLP and CNN than RNN in Fig 5?

I’d like to see some negative results where the variance is too high and harms training, to get a sense of how important it is to tailor the sampling strategies for different computational graphs.

Perhaps explain what M in section 5 is. I had to go to the appendix to figure out that it is a matrix that summarizes the discretization of the space and time.

Related work: Perhaps in the discussion of constant-memory via invertible transformation, Neural ODE can be included since it’s constant memory the same way as Gomez et al 17’.

--- post rebuttal --

Thanks for the response. I think the contribution of this work is solid and I vote for clear acceptance.

---

> ### Author Response · Authors · 2020-11-16
> **Response to reviewer #1**
>
> Thank you for your review, and we are glad you enjoyed the paper!
>
> Understanding the source of variance:
> - We agree it might be a good idea and we will look into adding some toy examples and quantitative analysis of variance in the final version of the paper. We have done empirical analysis on the variance, which was originally in Appendix A but is now in the main paper.
>
> Speculation on drop in performance in feedforward vs RNNs:
> - It is hard to say whether the performance really is more drastic since they are different models and problems. But if there is indeed a trend, it could be due to the fact that the MNIST and CNN experiment learning rate and weight decay were tuned for each gradient estimator separately, as described in section 4.3, while the RNN experiments had a fixed learning rate and weight decay for all the experiments (due to computational resource constraints).
>
> Negative examples:
> - Good idea. We will look at adding these to the next version of the paper. In our experience though, letting the variance become exponential in the depth of the network would cause the network to not converge at all (often the test accuracy would barely be above random). It is certainly very important in these cases to tailor the sampling strategies to the computational graph.
>
> Explaining M: We added some clarifications. Related work: Added the reference to Neural ODEs.

---

### Author Response · Authors · 2020-11-17
**New revision uploaded**

Dear reviewers,

Thank you for your valuable comments. We have uploaded a new revision with the various clarifications and nits discussed below. In particular, we clarified the unbiasedness of random matrix injection in 3.2, the discretization of the PDE in section 5, and the discussion of the two extreme examples in section 3.3. We also moved the empirical variance analysis from appendix A into the main paper.

Thank you,
The authors

---

### Comment · ~Ryota_Tomioka1 · 2021-02-22
**Question about Fig. 6**

Hi authors,

I have a question regarding the fairness of comparison with the "reduced batch" baseline.

If I understand correctly the reduced batch baseline sees only 20 examples per iterations compared to 150 examples in the large batch baseline. From a statistical point of view, doesn't it make sense to use the number of examples processed instead of number of iterations as the horizon axis?

(I understand that the focus of the paper is memory but this also means that "reduced batch" baseline requires 150/20-times less compute per iteration.)

Thanks!

---

> ### Author Response · Authors · 2021-02-25
> **Response to question**
>
> Thank you for your question!
>
> We are taking the viewpoint here that we want an unbiased approximation to the full gradient, which would ideally be the full dataset gradient, but we instead take as a "large" batch size of 150 since the baseline of full dataset gradient would be too expensive. Two ways of getting an unbiased approximation is to either sample from this "large" batch into a "reduced" batch, or to use RAD. We believe the comparison is fair in the sense that backpropagation takes a similar amount of memory for the two experiments. From a statistical point of view, since our datasets are small and we consider a large number of iterations, even in the "reduced batch" experiments the model sees all of the samples several times. In the case of CIFAR-10, as an example, the dataset size is 60,000 (note that we did not use data augmentation) . If our batch size is 20 the model will see the whole dataset in ~3,000 iterations. In figure 6, we show up to 100,000 iterations.
>
> It is true that the reduced batch will have a smaller computation cost. Theoretically RAD could have reduced computation cost with specialized implementations, but it is true that the reduced batch will still have fewer FLOPS.

---

### Comment · ~Yunjiang_Jiang1 · 2021-09-16
**naive questions about the practical implications**

Dear authors,

  This is an exciting work to someone like me who uses AD on a daily basis. But I have some doubts about its practicality. In a typical large NN, frameworks like tensorflow/pytorch do AD by leveraging the chain rule at every layer. The Bauer's "path integral" approach to AD seems only useful when aimed at computing a single gradient. Is it really a fair starting point for your memory efficiency claim?

  Another idea in reducing memory footprint is RevNet, which allows not storing gradient of the next layer during multi-layer transformer back-prop. This has been used in the Reformer network.  I look forward to hearing your thoughts!

---

### Decision · Program_Chairs · 2021-01-07
**Final Decision**

**Decision:**

Accept (Oral)

**Comment:**

The reviewers agree that this is an interesting and original paper that will be of interest to the ICLR community, and is likely to lead to follow up work.